# Gender Differences in Social Networks Based on Prevailing Kinship Norms in the Mosuo of China

**Siobhán M. Mattison** [1,2,*], **Neil G. MacLaren** [3], **Ruizhe Liu** [1], **Adam Z. Reynolds** [1], **Gabrielle D. Baca** [1], **Peter M. Mattison** [4], **Meng Zhang** [5], **Chun-Yi Sum** [6], **Mary K. Shenk** [7], **Tami Blumenfield** [1,8], **Christopher von Rueden** [9] **and Katherine Wander** [10]

1. Department of Anthropology, University of New Mexico, Albuquerque, NM 87131, USA; rzliu@unm.edu (R.L.); adamzreynolds@unm.edu (A.Z.R.); gabrielledbj@gmail.com (G.D.B.); tami.blumenfield@gmail.com (T.B.)
2. National Science Foundation, Alexandria, VA 22314, USA
3. Bernard M. and Ruth R. Bass Center for Leadership Studies, Binghamton University (SUNY), Binghamton, NY 13902, USA; nmaclar1@binghamton.edu
4. Department of Biology, University of New Mexico, Albuquerque, NM 87131, USA; pmatisson@unm.edu
5. Department of Cultural Heritage and Museology and Institute of Archaeological Science, Fudan University, Shanghai 200433, China; zhangmengwb@fudan.edu.cn
6. College of General Studies, Boston University, Boston, MA 02215, USA; cys@bu.edu
7. Department of Anthropology, Pennsylvania State University, State College, PA 16801, USA; mks74@psu.edu
8. School of Ethnology and Sociology, Yunnan University, Kunming 650106, China
9. Jepson School of Leadership Studies, University of Richmond, Richmond, VA 23173, USA; cvonrued@richmond.edu
10. Department of Anthropology, Binghamton University (SUNY), Binghamton, NY 13902, USA; katherinewander@binghamton.edu
* Correspondence: smattison@unm.edu

**Abstract:** Although cooperative social networks are considered key to human evolution, emphasis has usually been placed on the functions of men's cooperative networks. What do women's networks look like? Do they differ from men's networks and what does this suggest about evolutionarily inherited gender differences in reproductive and social strategies? In this paper, we test the 'universal gender differences' hypothesis positing gender-specific network structures against the 'gender reversal' hypothesis that posits that women's networks look more 'masculine' under matriliny. Specifically, we ask whether men's friendship networks are always larger than women's networks and we investigate measures of centrality by gender and descent system. To do so, we use tools from social network analysis and data on men's and women's friendship ties in matrilineal and patrilineal Mosuo communities. In tentative support of the gender reversal hypothesis, we find that women's friendship networks in matriliny are relatively large. Measures of centrality and generalized linear models otherwise reveal greater differences between communities than between men and women. The data and analyses we present are primarily descriptive given limitations of sample size and sampling strategy. Nonetheless, our results provide support for the flexible application of social relationships across genders and clearly challenge the predominant narrative of universal gender differences across space and time.

**Keywords:** social relationships; matriliny; patriliny; cooperation; evolution; behavioral ecology

## 1. Introduction

*In summary, we propose that men and women are equally social, but their sociality is directed differently. To caricature, female sociality is dyadic, whereas male sociality is tribal. In other words, men seek social connection in a broad group with multiple people, particularly by competing for a good position in a status hierarchy; women, in contrast, seek social connection in close personal relationships based on mutual, dyadic intimacy.* (Baumeister and Sommer 1997, p. 39)

Humans are a deeply cooperative species. Indeed, many have gone so far as to argue that humans cannot reproduce without the assistance of others (Hrdy 2009; Kramer 2010). Social relationships are sources of information, resources, and other support that promote reproductive success (Apicella et al. 2012; Page et al. 2017). Research suggests that men and women build, maintain, and leverage networks differently in ways that correspond with gender-specific reproductive and cooperative strategies (Benenson 2019; Seabright 2012). Prior work, however, in large part underrepresents low- and middle-income nations and participants from non-industrialized settings (see David-Barrett et al. 2015), and has predominantly relied on young children and adolescents rather than adult participants (see Vigil 2007). Thus, the generalizability of observed gender differences in the properties of social networks is potentially limited, as little attention has been focused on how gender differences in social relationships may vary across broader social contexts. This is despite well-characterized variation in population structure and household demography, which shape individuals' interactions and formation of social ties (Power and Ready 2019). In this paper, we compare gendered social networks in two villages—one matrilineal and one patrilineal—among Mosuo agriculturalists of Southwest China to test two hypotheses: gender differences in social network structure are universal versus gender differences in social structure are shaped by the social environment, including kinship.

Evolutionary hypotheses positing universal gender differences are based on the premise that male and female reproductive and cooperative strategies diverge (Geary 2006; Trivers 1972) and that social relationships therefore serve different purposes for men and women (e.g., Baumeister and Sommer 1997; Geary 2006). In particular, because females experience obligate parental investment in the form of pregnancy and lactation, women are generally expected to invest effort more intensively in relationships that enhance or do not conflict with childcare (Low 2005). By contrast, men are hypothesized to use social networks more frequently to achieve status-oriented objectives, and/or facilitate activities such as hunting, collective defense, or collective aggression (David-Barrett et al. 2015; Rose and Rudolph 2006; von Rueden et al. 2018). These hypothesized universal gender differences have generated the following specific predictions: 1. Men's social networks will be larger than women's networks and will include a higher proportion of casual and opportunistic 'weak' ties (e.g., Baumeister and Sommer 1997; Benenson 1990; Vigil 2007); 2. Higher-quality relationships in women's networks will manifest in more frequent communication ties, whereas higher-quality relationships in men's networks will be demonstrated through participation in joint activities (e.g., Pearce et al. 2021; Roberts and Dunbar 2015); 3. men will show a higher threshold for relationship conflict and will achieve post-conflict reconciliation sooner than women (e.g., Benenson et al. 2018; Benenson and Christakos 2003; Dunbar and Machin 2014); 4. men will preferentially socialize in larger and more hierarchically organized groups while women will gravitate toward more intimate, often 'dyadic' relationships (e.g., Baumeister and Sommer 1997; Benenson 2019; David-Barrett et al. 2015).

Taken together, these predictions, associated with what we call the 'universal gender differences' (UGD) hypothesis, have garnered a fair amount of support across a range of studies. For example, women have been reported to treat friends more like kin, and men to treat friends more like strangers and to pursue status-oriented relationship goals (Ackerman et al. 2007). In a study of American university students, men reported more friendships than women, and were more willing to sacrifice intimacy to secure more friends (Vigil 2007). In a study of American high school and middle school students, boys had more friendship connections than girls, while girls' networks showed more small-group clustering (Lindenlaub and Prummer 2021). In another university sample, Friebel et al. (2021) showed that women's friendships remained stable over time, whereas men's friendship connections were more flexible and opportunistic. In a large study of Facebook friendships, David-Barrett et al. (2015) found evidence consistent with women preferring same-sex dyadic relationships and men favoring larger, same-sex cliques. Similarly, an experience-sampling study of Dutch and American participants found that women engaged in collective activities in dyads more than men (Peperkoorn et al. 2020). In addition, studies

of same-sex groups suggest that boys and men tend to organize their groups hierarchically more than women, whereas women are more likely to enforce egalitarianism (Benenson and Abadzi 2020; Berdahl and Anderson 2005).

In a review of available evidence, Rose and Rudolph (2006) concluded that girls were more likely than boys to engage in prosocial relationships and were motivated by 'connection-oriented' goals, whereas boys had larger, more hierarchical networks, with 'dominance' and 'self-interest' goals at the fore. These differences are generally thought to solidify in adolescence (Benenson 1990). However, some studies observe mid- to late-age gender reversal. A study of Europeans found that men have more social contacts than women, particularly in young adulthood, but then this gender difference reverses in middle age as the number of contacts for both genders declines and as reproductive priorities shift (Bhattacharya et al. 2016). There are also studies that find no gender difference in network structure (e.g., Mengel 2020) or find a higher number of network contacts in women compared to men, such as in the context of an online game (Szell and Thurner 2013) or online communication more generally (Psylla et al. 2017).

Non-human primate studies are often used as evidence of the primacy of evolutionary differences between men and women, but such claims are likely over-generalized. For example, female baboons exhibit female-'typical' behavior, building relationships to benefit infant survival (Silk 2007), and chimpanzee and bonobos females build coalitions to defend kin and friends against male aggression (Newton-Fisher 2006; Tokuyama and Furuichi 2016). Male baboons, chimpanzees, and bonobos, by contrast, build relationships to compete for high rank and the mating opportunities it affords. Male philopatry is often invoked as a primary driver of differences in gender-specific networks, on the premise that philopatry was common, long enough, or universal during much of hominoid evolutionary history (e.g., Campbell 2013; David-Barrett et al. 2015; Vigil 2007). In chimpanzees, male social networks are based strongly on maternal kinship relationships (Mitani 2009), and, in gorillas, male philopatry results in more dispersed networks for males (Bradley et al. 2004), for example. However, philopatry is variable among non-human primates, where the benefits of social bonds to females have been argued to be a causal driver of female philopatry in a majority of non-human primate species (e.g., Wrangham 1980). Even in non-human primates most closely related to humans, where male philopatry prevails, it appears that the importance of female social bonds has been under-emphasized (Emery Thompson 2019) and that males and females show less divergent use of social bonds than sometimes reported (Langergraber et al. 2009; Psylla et al. 2017). Among humans, contemporary hunter-gatherers are highly flexible in residence (Kramer and Greaves 2011; Wood and Marlowe 2011). More generally, although male philopatry and female exogamy are modal among human societies (Murdock and White 1969), humans are remarkably flexible in post-marital residence (e.g., Surowiec et al. 2019), creating varying constraints on the social relationships available to men and women (Power and Ready 2019).

In general, conditions that decrease the differences between men's and women's reproductive capacities may result in less divergence in their use of social relationships to achieve reproductive goals. Monogamy, for example, limits men's reproductive success so that it more closely matches women's (e.g., Brown et al. 2009; Fortunato and Archetti 2010). In monogamous contexts, men's interests should be more closely (if not perfectly) aligned with their partners' interests and their social relationships more tightly focused on household concerns. A society's means of subsistence influences the divergence of men's and women's reproductive success (Holden et al. 2003; Low 2005; Mattison 2011). Forms of subsistence such as pastoralism generally support men's reproductive agendas because animals can be converted into reproductive success at higher rates for men than for women (for example, via polygynous unions). In such cases, coalition-building and status-enhancing relationships may enhance men's chances to attain disproportionate reproductive success. For example, men may accrue more exchange partners than women due to public displays of wealth, e.g., in highland New Guinea (Lederman 1990), resulting in increased status and mating opportunities. Other forms of subsistence, such as foraging

or horticulture, are less likely to produce surpluses that support disproportionate reproduction among men, while others such as offshore fishing may remove men from their households for long periods of time. In these cases, women often contribute significantly to household production, or run their households with less consistent involvement from men (Mattison et al. 2019). A relatively high degree of women's autonomy amidst a limited ability among men to convert resources to reproductive success should be associated with less divergence in men's and women's relationship-building strategies.

Because subsistence differences alter the possible divergence in men's and women's reproductive success, they have been tied to variation in kinship systems, whereby son-biased inheritance (patriliny) is favored when men's reproductive interests can be effectively supported, and daughter-biased inheritance (matriliny) when reproductive returns are greater via daughters than via sons (e.g., Cronk 2000; Fortunato 2012; Holden et al. 2003; Mattison 2011). Matriliny is also frequently, though not exclusively, associated with female philopatry (matrilocality; Fortunato 2019; Surowiec et al. 2019). Differences in the ecological conditions thought to give rise to matriliny are thus likely to alter the costs and benefits of gender-specific social strategies in ways that limit—and maybe reverse—the differences anticipated by the UGD. There are numerous pathways by which this might arise: first, men's absences in matrilineal systems may constrain the extent to which they can build large local networks; second, the relatively limited ability to convert resources to reproductive success may affect men's status differentials by limiting wealth differentials; and, third, women may adopt more 'masculine' social strategies as the opportunity costs of childcare are lessened by local kin support (i.e., increased presence of local allocarers). Such reversals are often deemed impossible in anthropology, where it is axiomatic that matriliny still involves men in authority (see Mattison et al. 2019). Yet, economic games in matrilineal communities have shown reversals in, e.g., risk taking (Andersen et al. 2008; Gong et al. 2015; Gong and Yang 2012), that support the view that matrilineal women may take on roles often assumed to be 'masculine'.

Forms of subsistence can also impact gender-specific social networks via their influence on culturally transmitted gender norms, particularly norms related to gendered divisions of labor. While highly variable, gendered divisions of labor tend to shunt women towards more intra-household labor and direct childcare, which has been argued to constrain childbearing women's socializing beyond the household (von Rueden et al. 2018). For example, women's group-level influence among the Mekranoti of the Brazilian Amazon was negatively associated with their parenting demands (Werner 1984), and among the Tsimané of the Bolivian Amazon, the number of women's but not men's cooperation partners was negatively associated with the number of dependent offspring (von Rueden et al. 2018). Women in many small-scale societies may tend to build larger social networks as they near menopause, in part due to fewer childcare demands (Brown and Kerns 1985). Subsistence practices may affect women's mobility and therefore their opportunities to form networks. For example, the transition to the plow may have made agricultural labor more strength-intensive and less compatible with childcare, thereby decreasing women's labor value outside of the home and decreasing their bargaining power in community-level politics (Alesina et al. 2013). Similarly, societies of historically pastoralist origins are more likely to promote norms restricting women's mobility and therefore their social influence (Becker 2019). Warfare can also promote greater gender differentiation in social networking, due to male coalition building (Rodseth 2012). In contrast, some local norms encourage women's work, and therefore social connections, outside the home. Among Shodagor fisher-traders in Bangladesh, for example, some women travel to rural villages to trade with ethnic Bengali women, whose religion restricts their interaction with unrelated men (Starkweather et al. 2020). These studies suggest that gender norms related to current and historical means of subsistence influence the relative sizes and natures of gendered social networks.

In this paper, we leverage variation in kinship norms and institutions among the Mosuo of Southwest China to investigate differences and similarities in gendered so-

cial networks. Evolutionary anthropologists have hypothesized that kinship systems are shaped by social and environmental circumstances (Alvard 2011; Shenk and Mattison 2011) in ways that alter the costs and benefits associated with gender-specific reproductive strategies (Holden et al. 2003; Mattison 2011; Mattison et al. 2016). We have argued previously that Mosuo matriliny is driven by limited reproductive differentiation between the genders due to a resource base (agriculture) that is expansive and not particularly productive and that does not therefore support a strongly divergent male reproductive agenda (see also Alesina et al. 2013; Brown et al. 2009), as well as norms and institutions that allow some men to limit investment in reproductive partners and parenting activities (Mattison 2011; Mattison et al. 2019; see also Fortunato 2012). By contrast, patriliny, which predominates in the Mosuo villages located in more rugged, mountainous terrain, appears to be reinforced by monogamous unions and the need for stable support from men of spouses and children (Mattison et al. 2021). This context presents an ideal test of the UGD hypothesis in humans—if men tend to pursue divergent strategies due to fundamental sex differences, which they inherit as part of humans' evolutionary legacy, then we would expect to see differences between men and women even among the matrilineal Mosuo. If typical gendered differences in social networks are due instead to flexible strategies that are sensitive to local socioecological circumstances affecting the costs and benefits of different social strategies, then we are unlikely to see the typical gendered differences among the Mosuo. We pay particular attention to matrilineal Mosuo women, who may, in their socioecological circumstances, build social ties in ways that are considered typically 'masculine' in the existing literature.

In a case study of two villages, one matrilineal and one patrilineal, we compare the universal gender differences hypothesis (UGD) and gender reversal hypothesis (GRH) with the following predictions: UGD: Men have larger networks (higher degree) across matriliny and patriliny; women's greater focus on intimacy and 'dyadic' relationships results in smaller networks (lower degree). GRH: Women will have larger networks (higher degree) than men in matriliny and men will have larger networks (higher degree) than women in patriliny. Further, we explore descriptive measures of centrality in social networks as evidence for or against consistent differences by gender (UGD).

## 2. Methods and Study Site

Population: The Mosuo are a population of roughly 40,000 agriculturalists residing in the Hengduan Mountains on the border of Sichuan and Yunnan Provinces in Southwest China. They are famous among anthropologists for their matrilineal traditions, involving inheritance that effectively moves through lineally related household women (Mattison 2011), prominent roles for grandmothers and maternal uncles (Shih 2010), and lack of consistent involvement in parenting by some, but not all, fathers (Mattison et al. 2014, 2019). Less well known are a geographically distinct population of Mosuo, who are patrilineal and whose norms involve transmission of wealth and status from parents to their sons, monogamous marriage, and more limited, if still relatively strong, autonomy for women (Mattison et al. 2021). Evidence suggests that the patrilineal Mosuo separated from the matrilineal region 500 years ago or earlier, establishing separate norms and institutions while continuing to identify as Mosuo and maintaining a variety of shared customs, language, religion, and attire (Mathieu 2003; Mattison et al. 2021). We have shown previously that these differences in kinship norms and institutions are associated with reversals in child gender preference (Mattison et al. 2016) and gender disparities in health (Reynolds et al. 2020). We speculated that some of this arises via more limited social support for women in patriliny (Reynolds et al. 2020), a pathway we begin to investigate here.

While little has been published about cooperation among the patrilineal Mosuo, ethnographic and quantitative evidence suggests that cooperation is extensive among the matrilineal Mosuo. Mosuo people routinely come together to help each other during planting and harvesting seasons, for example, and cooperate in the construction and repair of homes, preparations and costs of religious and cultural ceremonies, and joint economic

ventures (Shih 2010; Thomas et al. 2018). Large households help with domestic activities such as childcare, and household sisters are said to reproduce as a communal effort toward ensuring lineage and household longevity (Ji et al. 2013; Shih and Jenike 2002). At the same time, tourism and acculturation have led to an increasing fraction of households adopting non-normative institutions and plausibly acting more autonomously than might otherwise be expected (Blumenfield et al. 2018; Mattison 2010; Walsh 2001; see also Wu et al. 2015). The villages sampled in this study were both relatively far removed from sites of tourism and are locally considered to be relatively 'traditional'.

Data collection: We carried out social network interviews as part of the ENDOW project in an attempt to capture full networks of households in one matrilineal (N = 40 households) and one patrilineal (N = 30 households) community of Mosuo in the summers of 2017 and 2018. Accompanied by local guides, we walked from house to house and asked any available adult member of the household, man or woman, who could also comment on the networks and kin relatedness of other adult members of the household, to participate. We explained the study to potential participants and addressed their questions before obtaining their informed consent for the interview (UNM IRB 06915). We employed a name generator approach in which respondents were asked to free list individuals with whom they had various kinds of social ties (Marsden 2005). We focus here on a question that asked respondents to identify whom they considered close friends ('pengyou') with whom one would hang out or 'chat' ('liaotian') after dinner, a common activity among friends. Our goal was to obtain responses to interview questions for one adult man and one adult woman in each household. In cases where the opposite-gender respondent was not available at the time of interview, the main respondent answered in their stead. We did not note the names of any additional people present during the interview. Because data collection included a complete household census, we were able to infer the identity of the opposite gender respondent in most cases (with certainty if they were the only adult member of opposite gender residing in the household and with high confidence if the friendship network consisted mainly of similar-age peers). In some cases, additional cues were available, such as the respondent indicating that friends were 'my son's'. For cases where multiple opposite-gender individuals of similar age resided in the same household, we did not assign an identity for the second household respondent. A census of households allowed us to identify and verify individual-level attributes for egos, inferred egos, and alters who were present in the census; we also collected some individual-level data on egos and alters at the time of social network interview.

Data Analysis: The populations of interest in this study were all adults in two geographically distinct communities: 312 adults in the matrilineal area and 219 in the patrilineal area. In order to estimate the patterns of friendship ties in each location, we constructed social networks as follows. First, we included each individual who was either interviewed directly or whose friendship ties were identifiable from an interview, an ego, as a node in a location-specific network, regardless of gender. Second, we drew an undirected edge between each ego and any other individual (an alter)[1] named in response to the question, 'With whom do you/women/men hang out after dinner?' Multiple edges, indicating that both termini of the edge were egos and had nominated each other (6.0% of raw edges in the matrilineal location and 9.5% of raw edges in the patrilineal location), were treated as a single undirected edge. Although the results of the name-generating process could have been considered a directed network, undirected edges were used because not all nodes had the same opportunity to be associated with both in- and out-edges.

Networks were characterized according to standard metrics (Wasserman and Faust 1994). We used several measures of centrality: degree, the number of reported friendships for an individual, whether by that individual or by others; betweenness centrality, the extent to which a node fall on the geodesic paths of others in the network, which we normalized to (0, 1) with 0 indicating no shortest paths include the focal node and 1 indicating all paths do; and closeness centrality, the number of edges between a focal node and all other nodes in a network, which is undefined for disconnected nodes and

normalized to (0, 1) with higher values indicating relatively short distances to all other nodes. We also calculated transitivity, the number of a node's complete triads (a case in which three nodes were connected by edges between each of the three possible pairs) as a proportion of its connected triads (a case in which three nodes were connected by two edges such that one pair of nodes was not connected by an edge). Transitivity reflects clustering in a network—the extent to which friends tended to nominate the same individuals as friends. We further characterized the networks via density, the number of observed edges as a proportion of possible edges; centralization, the extent to which centrality is concentrated in a small number of nodes; network-level transitivity, calculated as the average of the node-level values; and the mean distance between nodes, calculated as the average length of all shortest paths in each network.

Our sampling procedure resulted in the exclusion of 57% of possible nodes in the matrilineal area and 58% in the patrilineal area—these individuals and any edges connecting to them are considered to be missing. Although these levels of missingness are relatively high, reasonable estimates of many network features may still be possible (Smith and Moody 2013). Furthermore, our strategy of interviewing heads of households and their closest household partners likely included individuals who were relatively more central to the overall network, potentially reducing the effects of missingness on estimation (Smith et al. 2017). Differences between the central tendencies of node-level statistics were tested with permutation tests; pairwise comparisons between categories of egos are reported here with equivalent analyses of all nodes reported in Supplementary Materials (SI). For each test, means were calculated for the contrasted sampled values (e.g., degree centrality for matrilineal men and women), where $\overline{X}$ was the larger mean and $\overline{Y}$ was the smaller mean. A *p*-value was calculated as follows. $X$ and $Y$ were concatenated, ordered randomly, divided into two new samples $X'$ and $Y'$ such that $n_{X'} = n_X$ and $n_{Y'} = n_Y$, where $n_{(\cdot)}$ represents the number of observations in the relevant data, and the difference $\overline{X}' - \overline{Y}'$ was calculated and stored. This procedure was repeated 10,000 times. The *p*-value was the proportion of simulated differences $\overline{X}' - \overline{Y}'$ that were more positive than the observed, empirical difference $\overline{X} - \overline{Y}$.

We constructed generalized linear models (GLMs) to test the primary hypotheses using the degree of each node as the outcome variable. Degree was approximately Poisson distributed and was not zero inflated. All nodes were considered in the GLM analysis, with controls included for whether the node was a primary respondent (interviewee), a secondary respondent, or not a respondent. These controls capture the fact that we did not interview all individuals represented by nodes and therefore not all nodes had the same chance to declare edges and were thus expected to have a lower observed degree.

All analyses were conducted in R (version 4.0.5) using the 'igraph' package (version 1.2.6, https://igraph.org, 22 May 2021); network visualization was done in Gephi (version 0.9.2, https://gephi.org, 25 May 2021).

## 3. Results

Descriptive Statistics: We interviewed 17 men and 23 women in the matrilineal area who provided information for an additional 15 men and 8 women, for a total of 32 men and 31 women. We interviewed 18 adult men and 12 adult women as primary respondents in the patrilineal area. In addition to their own information, these respondents provided information about 11 co-resident men and 17 co-resident women, for a total of 29 men and 29 women respondents (Table 1). We were unable to identify the second household respondent in 8 (20.0%) cases in matriliny and 2 (6.7%) cases in patriliny; we treated this information as missing. The full discovered network (all nodes) included 55 women and 78 men in the matrilineal community and 45 women and 47 men in the patrilineal community. The mean age of all nodes was 50.0 years versus 42.8 years for women versus men in the matrilineal community, and 46.8 years versus 47.8 years for women versus men in the patrilineal community. Mean years of education was slightly lower for both men and women in the matrilineal community.

**Table 1.** Sample characteristics.

| ALL RESPONDENTS | Matriliny | | Patriliny | |
|---|---|---|---|---|
| | Women | Men | Women | Men |
| N | 31 | 32 | 29 | 29 |
| *Individual characteristics: mean (standard deviation)* | | | | |
| Age (years) | 48.9 (9.1) | 45.9 (14.4) | 43.5 (11.4) | 47.4 (13.3) |
| Education (years) | 1.7 (3.5) | 4.3 (4.2) | 3.5 (4.2) | 5.2 (4.1) |
| *Network centralities: median (interquartile range)* | | | | |
| Degree | 4 (2, 6) | 3 (2, 4.25) | 2 (2, 4) | 3 (2, 4) |
| Betweenness | 0.012 (0.001, 0.037) | 0.016 (0.001, 0.039) | 0.017 (0.000, 0.063) | 0.019 (0.000, 0.066) |
| Closeness | 0.23 (0.21, 0.26) | 0.22 (0.19, 0.23) | 0.16 (0.14, 0.18) | 0.17 (0.14, 0.18) |
| Transitivity | 0.10 (0.00, 0.33) | 0.00 (0.00, 0.067) | 0.17 (0.00, 0.33) | 0.17 (0.00, 0.33) |
| ALL NETWORK NODES | Matriliny | | Patriliny | |
| | Women | Men | Women | Men |
| N | 55 | 78 | 45 | 47 |
| *Individual characteristics: mean (standard deviation)* | | | | |
| Age (years) | 50.0 (12.5) | 42.8 (14.5) | 46.8 (12.9) | 47.8 (12.9) |
| Education (years) | 2.3 (4.2) | 4.0 (4.1) | 2.9 (3.9) | 5.4 (4.0) |
| *Network centralities: median (interquartile range)* | | | | |
| Degree | 2 (1, 4.5) | 1 (1, 3) | 2 (1, 3) | 2 (1, 3.5) |
| Betweenness | 0.001 (0.000, 0.015) | 0.000 (0.000, 0.013) | 0.000 (0.000, 0.034) | 0.000 (0.000, 0.031) |
| Closeness | 0.24 (0.21, 0.26) | 0.23 (0.20, 0.26) | 0.16 (0.15, 0.18) | 0.16 (0.14, 0.17) |
| Transitivity | 0.23 (0.00, 0.63) | 0.00 (0.00, 0.00) | 0.17 (0.00, 0.33) | 0.17 (0.00, 0.33) |

Qualitatively, the matrilineal friendship network appeared to be more connected and contained many more cross-gender friendships than the patrilineal network (Figure 1, left). By contrast, the patrilineal friendship network (Figure 1, right) was markedly gender segregated, with distinct clusters of friends, linked loosely to each other. In the patrilineal network, 2 (1.8%) discovered edges were cross-gender; in the matrilineal network, 24 (14.0%) discovered edges were cross-gender. There were five isolates reporting no friendships in the matrilineal area and three isolates in the patrilineal area.

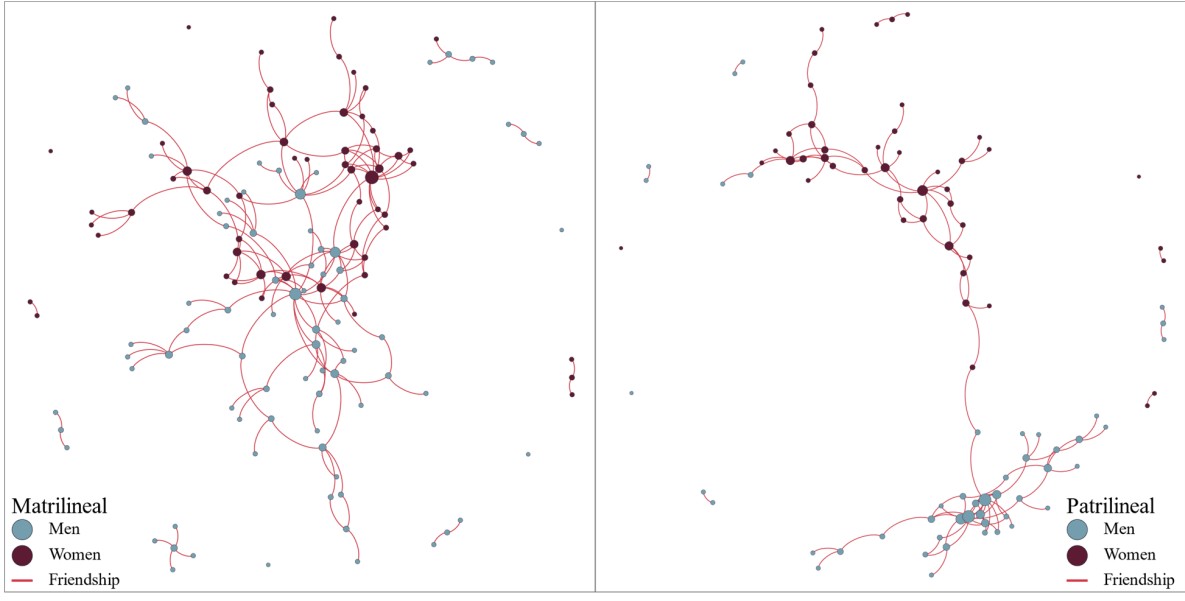

**Figure 1.** Social networks of men (blue) and women (purple) in matriliny (**left**) and patriliny (**right**).

Among respondents, the degree distribution of matrilineal women was distinct from the others (Figure 2A; Table 2); particularly, matrilineal women had a higher mean degree than matrilineal men ($p = 0.019$) and patrilineal women ($p = 0.069$). Men's degree distributions were roughly similar, with long right tails, which is consistent with a few men having relatively more friendships than the bulk of men and women. The degree distribution for matrilineal women stood out as being relatively flat, with a higher mean and median reported degree. The modal degree for respondents of both genders in both contexts was two to three. The direction of differences was similar when all nodes were considered (Table 1; SI Figure S1a).

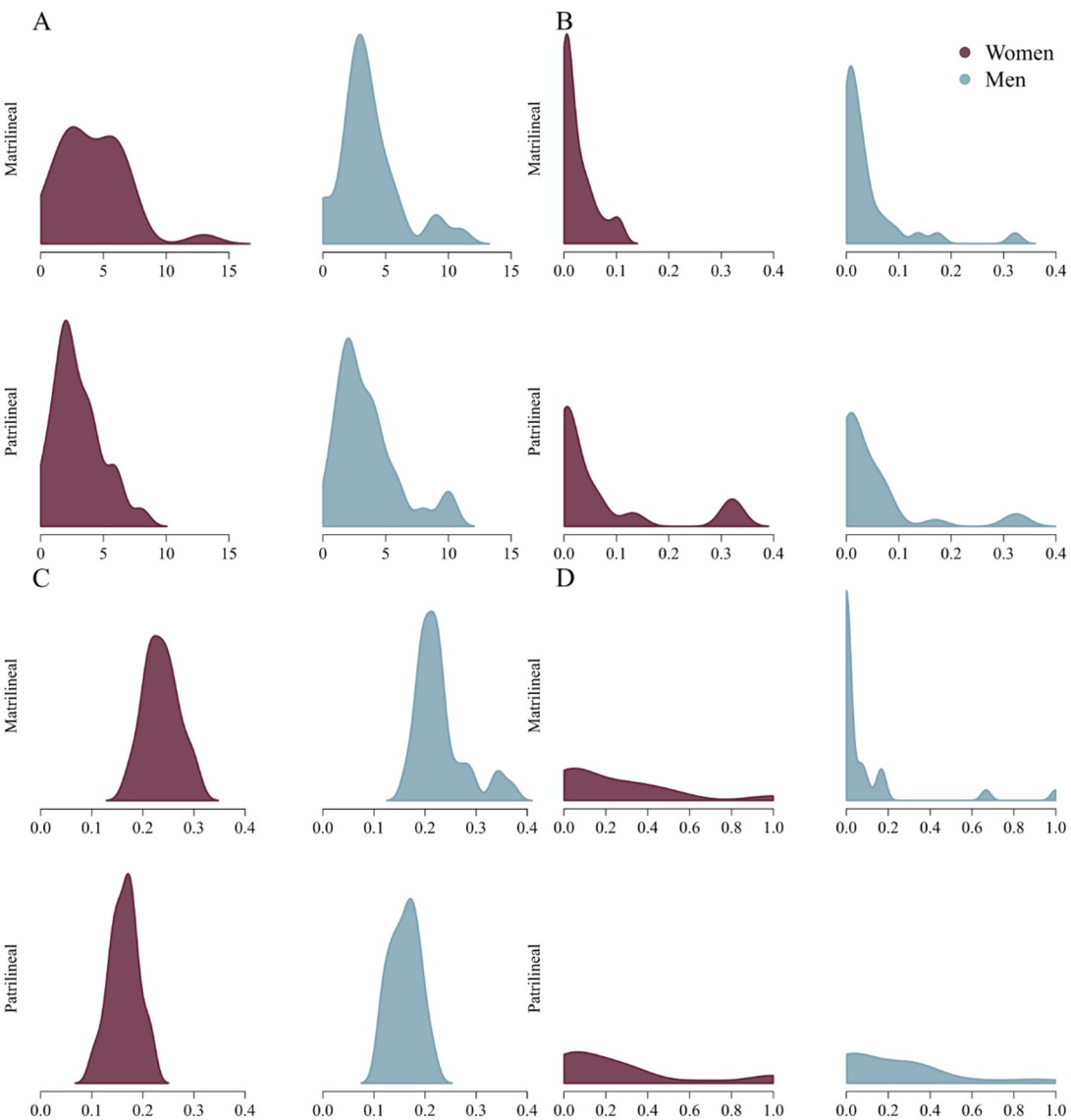

**Figure 2.** Kernel density plots for women (purple) and men (blue) in matriliny and patriliny, as labeled for (**A**) degree, (**B**) betweenness centrality, (**C**) closeness centrality, and (**D**) transitivity.

**Table 2.** Results of permutation tests comparing degree, betweenness, closeness, and transitivity distributions for patrilineal (pat.) and matrilineal (mat.) Mosuo respondents.

| Distributions Being Compared | | Mean 1 | Mean 2 | *p* |
|---|---|---|---|---|
| *Degree* | | | | |
| **Mat. Women** | **Mat. Men** | **3.018** | **2.205** | **0.019 *** |
| Pat. Men | Pat. Women | 2.596 | 2.356 | 0.315 |
| Pat. Men | Mat. Men | 2.596 | 2.205 | 0.170 |
| **Mat. Women** | **Pat. Women** | **3.018** | **2.356** | **0.069 †** |
| *Betweenness* | | | | |
| Mat. Men | Mat. Women | 0.017 | 0.014 | 0.351 |
| Pat. Women | Pat. Men | 0.045 | 0.031 | 0.214 |
| **Pat. Men** | **Mat. Men** | **0.031** | **0.017** | **0.096 †** |
| **Pat. Women** | **Mat. Women** | **0.045** | **0.014** | **0.009 *** |
| *Closeness* | | | | |
| Mat. Women | Mat. Men | 0.239 | 0.233 | 0.224 |
| Pat. Women | Pat. Men | 0.163 | 0.158 | 0.221 |
| **Mat. Men** | **Pat. Men** | **0.233** | **0.158** | **0.000 *** |
| **Mat. Women** | **Pat. Women** | **0.239** | **0.163** | **0.000 *** |
| *Transitivity* | | | | |
| **Mat. Women** | **Mat. Men** | **0.368** | **0.067** | **0.000 *** |
| Pat. Women | Pat. Men | 0.258 | 0.247 | 0.498 |
| **Pat. Men** | **Mat. Men** | **0.247** | **0.067** | **0.001 *** |
| Mat. Women | Pat. Women | 0.368 | 0.258 | 0.211 |

† denotes significance at α = 0.1. * denotes significance at α = 0.05.

Additional measures of centrality revealed few strong differences between men and women. In betweenness centrality, the importance of one cross-gender edge in bridging the largely gendered clusters in the patrilineal community was apparent (Figure 2B); nodes along this path had correspondingly high levels of betweenness centrality. Matrilineal women demonstrated low levels of betweenness centrality, potentially suggesting matrilineal women were generally more connected within that network. Patterns in betweenness centrality were similar when all nodes were considered (Table 1; SI Figure S1b).

Under matriliny, the range in closeness centrality was wider for men than for women, although the means of the distributions were not obviously different (Figure 2C; Table 2). Among respondents from the patrilineal community, closeness centrality distributions were similar between men and women. There were no statistically significant differences in closeness by gender within either community; however, higher closeness in the matrilineal community was apparent upon both visual inspection (Figure 2C) and pairwise tests (Table 2). This was consistent with higher connectivity in the matrilineal friendship network and increased average distance between nodes in the patrilineal network due to the single set of edges connecting the two main clusters of nodes. Consideration of all nodes does not alter this conclusion (Table 1; SI Figure S1c).

The modal transitivity for both women and men respondents in both matriliny and patriliny was zero. Matrilineal women had a higher transitivity than matrilineal men ($p < 0.001$). The mean transitivity for men and women in patriliny was similar. Patterns were similar when all nodes were considered (Table 1; SI Figure S1d).

At the network level (SI Table S1), network density, degree centralization, and betweenness centralization were roughly equivalent between the two networks, while measures of network distance diverged. In particular, although all nodes were on average closer to each other in the matrilineal network, there was more concentration of closeness centrality in the matrilineal network than in the patrilineal network.

Generalized Linear Models: Modeling degree as a function of gender and community (matrilineal or patrilineal), with controls for respondent status (i.e., primary respondent, inferred respondent, or discovered node) and age did not reveal differences by gender (Coef for men: −0.04; 95% CI: −0.21, 0.12; Table 3). There was marginal evidence for a positive association between matriliny and degree (Coef: 0.15; 95% CI: −0.02, 0.32). The

interaction between gender and community predicted by the gender reversal hypothesis was not apparent (Coef: −0.21; 95% CI: −0.56, 0.13). However, differences by gender across community were in the predicted direction: in matriliny, men had a lower degree (Coef: −0.13, 95%CI: −0.35, 0.09), while in patriliny, men had a higher degree (Coef: 0.09, 95%CI: −0.18, 0.35). The control variable for survey respondent was, as expected, positively associated with degree, as respondents had more edges than non-respondents; the control variable for inferred respondent was inversely associated with degree, suggesting fewer ties may have been reported for secondary respondents within a household. Predicted probabilities from this model supported a mild gender reversal in degree by descent system, providing tentative, suggestive evidence for a gender reversal in network size (Figure 3).

**Table 3.** Poisson generalized linear models of network degree by primary household survey respondent status, secondary household survey respondent status, age, gender, and community for all nodes of matrilineal and patrilineal Mosuo adults (n = 225).

| Variable | Coefficient | Standard Error | 95% CI | *p* |
|---|---|---|---|---|
| Intercept | 0.04 | 0.20 | (−0.35, 0.42) | 0.84 |
| **Control for primary respondent** | **1.11** | **0.11** | **(0.90, 1.33)** | **<0.0001 \*** |
| Control for secondary respondent | −0.19 | 0.10 | (−0.39, 0.01) | 0.065 † |
| Age | 0.00 | 0.00 | (−0.00, 0.01) | 0.37 |
| Gender (man) | −0.04 | 0.09 | (−0.21, 0.12) | 0.62 |
| **Community (matrilineal)** | **0.15** | **0.09** | **(−0.02, 0.32)** | **0.095 †** |
| Intercept | 0.00 | 0.20 | (−0.40, 0.40) | 1.0 |
| **Control for primary respondent** | **1.10** | **0.11** | **(0.89, 1.31)** | **<0.0001 \*** |
| Control for secondary respondent | −0.17 | 0.10 | (−0.37, 0.03) | 0.11 |
| Age | 0.00 | 0.00 | (−0.00, 0.00) | 0.45 |
| Gender (man) | 0.09 | 0.14 | (−0.18, 0.35) | 0.53 |
| **Community (matrilineal)** | **0.26** | **0.13** | **(0.01, 0.52)** | **0.04 \*** |
| Gender (man) * Community (matrilineal) | −0.21 | 0.18 | (−0.56, 0.13) | 0.23 |

† denotes significance at α = 0.1. * denotes significance at α = 0.05.

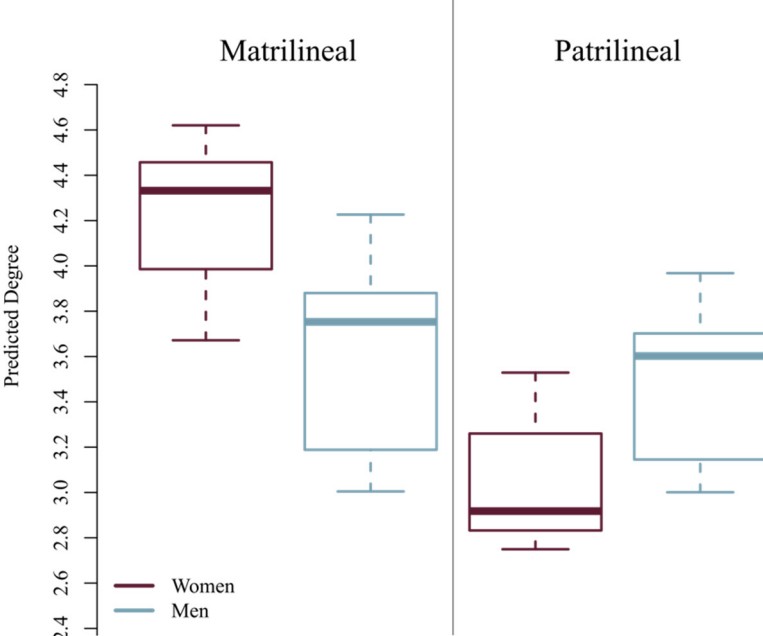

**Figure 3.** Predicted probabilities for network degree by gender and descent system controlling for survey respondent status and age.

## 4. Discussion

In this paper, we strove to test what we have called the universal gender differences (UGD) hypothesis, which posits, among other things, that men have larger networks than

women, here operationalized as degree in an undirected social network. We tested this in two communities—one patrilineal and one matrilineal—that are organized differently in terms of gender, inheritance, descent, and ecology, but are otherwise similar (Mattison et al. 2021). This variation presented the opportunity to investigate whether the gender differences of the UGD hypothesis might hold in some ecologies, but not others, and whether gender reversals in node degree (a measure of network size) might be present in matrilineal communities where women are relatively autonomous (Mattison et al. 2019; Reynolds et al. 2020).

We find no evidence in either patriliny or matriliny in support of the UGD hypothesis. In patriliny, there are no marked gender differences in network metrics. In matriliny, there are some gender differences: although these differences are not substantial, they are generally in the opposite direction to predictions of the UGD hypothesis. Specifically, matrilineal women may have larger networks than matrilineal men. Additionally, both descriptive results and our generalized linear model suggest that matrilineal friendship networks are larger and more connected than patrilineal networks, with matrilineal friendship networks including more hetero-gendered social relationships and patrilineal ones clustering more strongly on gendered lines. In addition to rejecting the UGD in these contexts, we interpret this evidence as consistent with variation being structured by local socio-ecologies. These inferences are tempered by limitations of sample size (only two networks were sampled) and sampling strategy (not all relevant individuals were interviewed). Nonetheless, we believe this is a useful case study if it prompts additional research in small-scale settings where local gender norms and ecological variation may be likely to reveal additional nuances in the ways that men and women structure and use social support.

These findings suggest that local socioecological circumstances should be incorporated into understandings of women's and men's constraints, decision making, and associated outcomes (Winterhalder and Smith 2000). More specifically, the assumptions that underlie hypotheses of universal differences in men's and women's social and reproductive strategies are unlikely to be met across contexts. We highlight two such assumptions here, as neither is likely to characterize both subpopulations of the Mosuo. First, the UGD hypothesis assumes that reproductive variance is higher in men than in women and that this motivates men to cultivate larger ('diffuse') networks as a means to gain status (and, as a result, more mating opportunities). Furthermore, sexually selected motivations contribute to gendered divisions of labor, and these tend to anticipate more intra-household labor by women, which may, under some circumstances, constrain women's social-networking opportunities relative to men's (von Rueden et al. 2018). Among the patrilineal Mosuo, monogamy and the consistent need for men's labor (Mattison et al. 2021) likely constrains men who would otherwise attempt to pursue multiple reproductive partners (see, e.g., Fortunato and Archetti 2010; Kokko and Jennions 2008). More generally, although variance in reproductive success is higher, on average, for men than for women among humans, there is substantial population-level variation (Brown et al. 2009), with variance in female reproductive success occasionally outstripping male. Monogamy characterizes the bulk of relationships among both patrilineal and matrilineal Mosuo (Mattison et al. 2021), but exists within the context of relatively small, mostly nuclear domestic units in patriliny, suggesting that status-oriented pursuits may have more limited impact on male reproductive success among the Mosuo—even in patriliny—than it might in contexts where polygyny prevails.

Second, the UGD hypothesis often references male philopatry and female emigration as an ancestral pattern of community organization to which men and women adapted differently. Specifically, scholars have argued that general tendencies toward male-biased philopatry in humans should lead to stronger, kin-based coalitionary networks in men versus a focus on fewer, more intimate relationships in women (Vigil 2007; Wrangham 1980). Although patrilocality is the modal form of human social organization in contemporary societies (Murdock and White 1969), it is far from universal (Surowiec et al. 2019) and is unlikely to have been the single emigration pattern of Pleistocene foragers, who most likely

displayed flexibility in residence (Dyble et al. 2015; Kramer and Greaves 2011; Wood and Marlowe 2011)—and in fact it has been argued that modern low levels of matriliny and high levels of patriliny may be in part due to the impact of colonialism on the ethnographic record (Shenk et al. 2019). In our study communities, patrilineal women initially have more limited access to social support than patrilineal men. By contrast, matrilineal women and men normatively reside in their natal communities throughout their lifetimes. Our findings provide preliminary evidence that matrilineal friendship networks are larger than patrilineal ones and are also more likely to involve hetero-gendered relationships. This pattern of natalocal residence may contribute to that (He et al. 2016; Shih 1993), as may the relative ease with which individuals travel from house-to-house in this relatively traversable terrain (Mathieu 2003; Mattison et al. 2021). This result, even if tentative, reinforces the importance of differences in the social and demographic constraints imposed by kinship systems in structuring access to social support and the costs and benefits to men and women of gender-specific strategies (Koster et al. 2019; Low 2005; Power and Ready 2019; Starkweather et al. 2020), with important implications for understanding strategies across diverse contemporary settings (David-Barrett 2019; Mattison and Sear 2016).

Our qualitative assessment of degree distributions suggests differences between genders, by descent system, offering very tentative support for our hypothesis that the 'typical' pattern of a higher degree for men would be reversed in the matrilineal setting. Our analyses did not bear out the predicted interaction between gender and community, potentially due to our fairly small sample size. Although our qualitative assessment hardly stands on its own, in the context of other studies showing gender reversals in aspects of behavior (Gong et al. 2015; Gong and Yang 2012; Liu and Zuo 2019) and outcomes (Mattison et al. 2016; Reynolds et al. 2020) among matrilineal Mosuo relative to other patrilineal populations, we suspect that future work in a larger sample may bear out this prediction.

Our findings provide additional motivation to reexamine existing dogma stipulating the absence of matriarchy and limits to women's roles in matriliny (see Leonetti et al. 2005; Parkin 2021; Schneider 1961). Women in matriliny are often implicitly or explicitly assumed to never fully take on roles analogous to those of men in patriliny, particularly as societal leaders. However, a variety of evidence points to the importance of women's leadership and status in securing evolutionarily relevant benefits, such as their own and their children's health and welfare, even if women's influence on average in the range of societies that generate the ethnographic corpus may be less overt than men's (Alami et al. 2020; Reynolds et al. 2020). Matrilineal women's social relationships as depicted here do not fully mirror men's; still, their relatively large network size suggests that aspects of women's social strategies can resemble men's when women are more central and have more authority and social support (Hrdy 2000; Mattison et al. 2019; Smuts 1994). The large households characteristic of the matrilineal Mosuo are likely to free many women from the demands of childcare by providing ample and capable allocarers, whereas the nuclear households characteristic of patrilineal Mosuo are likely to be conducive to more domestic-oriented activities among women, as suggested by our ethnographic observations. In other small-scale societies, women's time spent in childcare and domestic labor more generally can contribute to greater opportunity for men to socialize broadly within and between communities and to gain community-wide social influence (Werner 1984; Brown and Kerns 1985; von Rueden et al. 2018). Differences in network characteristics according to social and ecological contexts underscore the importance of general, rather than gender-specific, models of reproductive and cooperative strategies that consider social, demographic, and ecological constraints affecting the potential for complementarity, supplementarity, and divergence between men and women (Bliege Bird and Bird 2008; Mattison 2016; Starkweather et al. 2020).

Our study is subject to a number of important limitations. Our sampling strategy resulted in a network that is not complete in that we did not interview every person residing in the two communities, but rather members of, to the best of our ability, every household in each community. Households often included more adults than the two about

whom friendship information was sought. Our expectation is that if more individuals had been interviewed, the number of nodes and edges would have increased as well, affecting, for example, the maximum observed degree. Furthermore, although node degree and transitivity in networks of a similar number of nodes as our potential networks (250–350 nodes) is relatively consistently estimated across a wide range of percent missingness, metrics such as betweenness and closeness are more sensitive (Smith and Moody 2013). This sensitivity may be problematic in the networks we studied because betweenness and closeness are driven in large part by the small number of cross-gender friendship ties, particularly in the patrilineal network. However, the difference in proportion of cross-gender edges between the two locations is stark: although our precise estimates of, for example, betweenness and closeness may differ from values that would have been obtained in the complete network, cross-gender ties would have to be substantially undersampled in the patrilineal network to approach the proportion of cross-gender ties in the matrilineal network. That said, a sampling strategy that focused on younger or older adults, or a more random sample of individuals (and thus less systematically likely to include higher centrality nodes (Smith et al. 2017)), may have found different patterns. Moreover, this is a static, descriptive portrait of social relationships, which are known to change across the life course (David-Barrett et al. 2016; Palchykov et al. 2012; Bhattacharya et al. 2016). Stability of relationships may have important implications for health, well-being, and reproductive success (Cheney et al. 2016) that we cannot capture here, perhaps particularly for women in patriliny where many relationships are established subsequent to marriage. Our ability to capture spatially and temporally diverse friendships that characterize humans and distinguish us from non-humans (Rodseth et al. 1991) is incomplete. Nor do we have information on the intimacy or specific exchanges implied by observed relationships that might help to test differences in the quality and intensity of relationships anticipated by some of the hypotheses reviewed in the introduction (e.g., Rose and Rudolph 2006; Vigil 2007). This, too, would provide important insights on how the strength of relationships relates to fitness and well-being (Scelza 2011; Silk et al. 2010), and whether this varies by gender across different social systems. Finally, we characterized networks based on self-reported, often unreciprocated ties. Although we have no reason to suspect biases in any particular direction, we also have no observation of benefits transferred along the reported ties. Observations of specific types of cooperation (e.g., working on someone's farm (Thomas et al. 2018)) would help to verify the importance of the patterns we describe here.

Certainly, the fact that respondents' social networks are larger than others' suggests the need for more complete sampling. Comparing our results to prior network studies based on more complete networks drawn from other settings, e.g., classrooms (Benenson 1990; Vigil 2007) or even relatively clearly bounded networks (Apicella et al. 2012; Nolin 2012; Page et al. 2017), is difficult. Considering the limitations of network methods and metrics in field settings, especially with loose boundaries among communities (Gerkey and Cronk 2014), will be important in future studies of gender differences in social relationships. Future comparative networks studies that build upon these results will facilitate attempts to generalize findings.

## 5. Conclusions

Despite the demonstrated importance of social relationships to human and non-human primate reproductive success and well-being, there has been remarkably little work in evolutionary anthropology investigating how social relationships may be structured and used differently by men and women in varying contexts. Our study is the first of which we are aware to compare men's and women's social networks in two very different kinship contexts: matriliny and patriliny. Because these contexts are part of the same culture yet differ in the extent to which they support divergent reproductive strategies, they are ideal for understanding how fixed (or flexible) gendered social relationships are. Women provide important forms of support that often go unrecognized in evolutionary

studies of cooperative networks. We have shown here that gender differences in social network size can reverse in matriliny compared to patriliny. This suggests the need to evaluate common assumptions undergirding universal models of gender differences in social and reproductive strategies, which are only likely to be met in some socioecological circumstances, and which were unlikely to have fully characterized ancestral human environments. Our point is not to say that men and women never diverge or that they do not pursue complementary strategies, or that the patterns found by previous researchers are incorrect, but rather to suggest that more general models of human evolutionary strategies that incorporate non-gender-specific constraints and consider diverse socioecologies will offer a broader understanding of human flexibility. Given the links between social support, health, and well-being across species (Cheney et al. 2016; Power and Ready 2019; Silk et al. 2003), this is not merely a theoretical exercise.

**Supplementary Materials:** The following are available online at https://www.mdpi.com/article/10.3390/socsci10070253/s1, Figure S1: Kernel density plots for men (blue) and women (purple) in matriliny and patriliny, Table S1: Network-level density, centralization, transitivity, and distance statistics for Mosuo matrilineal and patrilineal networks.

**Author Contributions:** Conceptualization, S.M.M.; methodology, N.G.M.; formal analysis, N.G.M.; investigation, S.M.M., A.Z.R., M.Z., C.-Y.S.; data curation, S.M.M., N.G.M., R.L., A.Z.R., P.M.M., C.-Y.S., K.W.; writing—original draft preparation, S.M.M., N.G.M., R.L., A.Z.R., K.W.; writing—review and editing, S.M.M., N.G.M., R.L., A.Z.R., G.D.B., P.M.M., M.Z., C.-Y.S., M.K.S., T.B., C.v.R., K.W.; supervision, S.M.M., K.W.; project administration, S.M.M., C.-Y.S., M.K.S., T.B., K.W.; funding acquisition, S.M.M., M.K.S., T.B. All authors have read and agreed to the published version of the manuscript.

**Funding:** The work was funded by the National Science Foundation (BCS 1461514 & 1461520). This material is based upon work supported by (while serving at) the National Science Foundation. Any opinions, findings, and conclusions or recommendations expressed in this material are those of the author(s) and do not necessarily reflect the views of the National Science Foundation.

**Institutional Review Board Statement:** The study was approved by the Institutional Review Board of the University of New Mexico (06905, effective 19 May 2021).

**Informed Consent Statement:** Informed consent was obtained from all subjects involved in the study.

**Data Availability Statement:** Data are available via the lead author's Github repository: https://github.com/smattison/Social-Sciences-Data (accessed on 30 June 2021).

**Acknowledgments:** Thanks first and foremost to study participants who generously gave their time and insights to help us with our work. Three reviewers, the editors, and Jeremy Koster provided valuable criticisms of the work. The work was funded by the National Science Foundation (BCS 1461514 & 1461520). This material is based upon work supported by (while serving at) the National Science Foundation. Any opinions, findings, and conclusions or recommendations expressed in this material are those of the author(s) and do not necessarily reflect the views of the National Science Foundation.

**Conflicts of Interest:** The authors declare no conflict of interest.

## Notes

[1] This modification to standard use of the terms ego and alter is used in this study to reflect the fact that our sampling strategy resulted in individuals having systematically more or less opportunity to have their friendship edges represented in the network.

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
