# Peer review of "Gender Differences in Social Networks Based on Prevailing Kinship Norms in the Mosuo of China"

_socsci, doi:10.3390/socsci10070253_

Round 1

Reviewer 1 Report

SOCIAL SCIENCES

Gender reversals in social networks based on prevailing kinship norms in the Mosuo of China

This study is a concise, well-illustrated overview concerning the relevance of studying gender differences of social network structures. The findings among matrilineal and patrilineal Mosuo communities are consistent with existing studies that document gendered network patterns. This human behavioral ecology (HBE) study is original in its application of social network analysis (SNA) to explore gendered networks in different kinship systems. I felt that the authors introduced the topic effectively and aptly documented the different network patterns they found through clear prose. I have no suggestions for improvement.

Author Response

Thank you very much for your comments.

Reviewer 2 Report

Review comments for Gender reversals in social networks based on prevailing kinship norms in the Mosuo of China

General comments:

Fantastic work. It is deals with an important question, and approaches it brilliantly. It is actively trying to push out human knowledge. Wow.

The sex difference in social network building strategies may be key to how modern societies build their social networks.

For the non-sex specific version, see for this:

David-Barrett, T. (2020) Herding Friends in Similarity-Based Architecture of Social Networks. Scientific Reports 10, 4859. doi: 10.1038/s41598-020-61330-6

David-Barrett, T. (2019) Network Effects of Demographic Transition, Scientific Reports 9:2361, doi.org/10.1038/s41598-019-39025-4

(Reading this paper made me feel that I what is missing is a gendered version of these papers that would suggest empirically testable network measure hypotheses.)

Detailed comments:

page 2

“However, this research rarely takes an explicitly evolutionary perspective; furthermore, it has left low- and middle-income nations and non-Western participants under-represented (David-Barrett et al., 2015) ”

Er… we had an explicitly evo approach. And had participants from all around the world, including many non-Western countries. It is true though that all the data was coming from Facebook users, which at the time of data collection was more middle class than it is today. (We repeated the study watching people on the streets all around the world, which has an even larger sample – well above a million people. The finding was eerily the same.)

"Men are hypothesized to use social networks to achieve status-oriented objectives, whereas women’s networks are hypothesized to be geared toward affiliative activities, such as childcare and other domestic activities"

This is a little unfair to the literature... It reads as if the past has been all a bunch of sexist assumptions. This was not the case in the David-Barrett et al 2015 paper.

page 3

"David-Barrett et al. (2015) found evidence consistent with women favoring dyadic relationships and men favoring larger, all-male cliques"

To be precise here: what we found was that women preferred same-sex dyadic relationships compared to men. (It is correctly stated that the large groups were either all male, or almost all male.)

Line 87. The para starting with “Many evolutionary arguments appeal…”

We always scratched our heads about this… There are primates that are not patrilocal, and the modern-day foragers are super flexible both within and across cultures. Yet, both bonobos and chimps are patrilocal. So, in our search for a possible explanation for what we found in the FB profile pic dataset, we ended up with thinking that a long phase of human evolution could have had patrilocality, giving enough time for a sister-like best friend behaviour template to evolve.

(I should add that we tested a host of alternative hypotheses to why the 2F/2M profile picture frequency ratio was well above 1 everywhere, but I stupidly messed up the ethics for these, so we could not publish them. We found that the effect was not coming from male fear of homophobia, relatively difference in female homosexual v hetero behaviour, and the simple preference for dyads of any sex.)

Lines from 126: The hypotheses

The least useful reviewer comment is to point out that n=1 in this dataset, given that there is only 1 matriliny and 1 patriliny under observation. Because of this we cannot possibly know if the effect hypothesized is coming from the factor that drives whether the kinship system is matri- or patri-lineal, or whether the kinship system is itself at work. In other words, are the terrain and the technology driving the network size and the kin/friend ratio in the network, or rather the consequent kinship system?

Although of course it is impossible to solve this trouble with this data (and I cannot really see another dataset out there that would do so), acknowledging the fact that the problem is there would make the paper stronger.

Lines 166-184. Concerning the data collection methods.

This section stays a little abstract: mentions the layers, and some of the terms, but does not go into detail. I would find it useful (and I presume so would all the fellow network nerds) to read the methods in detail, i.e., exact steps, exact question.

Lines 185-200.

The same holds for the data analysis and the ERGM section. I would love to read more detail.

Line 201. Results section

I suggest reworking the entire results section. It carries two problems.

  1. It would be helpful to give us general descriptions of the social networks that you found. I had trouble seeing the networks in front of my eyes, and thus having a perception of what might be going on other than what you suggested.
  2. I have had real trouble following how the calculations exactly happened. Could you please give much more detail?
  3. Many of the analysis statements were not accompanied by statistical analysis. It is impossible to tell to what extent the verbal statements about them are correct.
  4. The verbal analysis would benefit from much more detailed and structured treatment.

I suggest that after you do these changes, the referees look at it again. In this current state I cannot be helpful apart from pointing out the need for more detail. (If there is a Supplementary Material file, it would be useful to have a look. Will you upload the raw data? If so, please include that in the next round of revisions.)

Lines from 256: Discussion

A great write up.

It would be useful though not to overgeneralise the results. This study had n=1. Perhaps it would be helpful to acknowledge this, and say something like this: if this pattern holds in general….

Line 339:

Another paper that tracks this, including life-course dependent social network edges in different relationship types is:

David-Barrett, T., J. Kertesz, A. Rotkirch, A. Ghosh, K. Bhattarcharya, D. Monsivais, K. Kaski, (2016) Communication with family and friends across the life course, PLOS ONE  11:11

Not surprisingly the extent the edge changes across the life course is dependent on the type of the relationship. (Which may have a consequence for your research methods.) NB. We repeated the same method on another dataset (3bn calls), one in which we could validate rather than infer the relationship types with the same finding.

Line 354:

It is exactly for this reason why it would be helpful to know more about the graphs. In their current form I can not judge what is a valid network statement and what is not.

Line 365: a word is missing?

Line 366:

“We have shown here that, across a number of important metrics, gender differences in social and cooperative networks reverse in matriliny compared to patriliny. ”

This statement feels unjustified. Instead: the research here demonstrated that such a reversal can happen.

Reviewer 3 Report

Thank you for the invitation to review this paper. Overall, I like the aims and perspective of the manuscript and believe is offers an important contribution to the literature. The authors leverage of the diversity in kinship structures in the Mosuo to explore how gendered social network changed within the same population but under different kinship norms. By doing so they are able to highlight that the previous assumption of gendered differences in network structures (males = large, kin based and females = small, dyadic and more intimate (whatever intimate means – seems highly loaded to me!)) do not hold when we look outside of western contexts and male philopatry/patrilineal systems. As the authors point out, leveraging on diversity to specific test hypotheses like this is a real strength of behavioural ecology which is this a great example of.

While overall I think this paper has a lot of potential, I do have a number of suggestions, which I think, will strengthen the paper. My concern is, at the moment, is that the findings of the paper will have less influence without some additional explanation and alterative approaches. Generally, I did find the methodological section lacking so if I have misunderstood that the study was conducted (i.e. I suggest something you have actually done!) then this might be indicative of the need to expand the data analysis description further. As a fair warning, I tend to write a lot in reviews – this isn’t I hope demoralising but really just me trying to help.

I will cover first some overarching comments / more major suggestions and then give a bit more detail for some minor suggestions.

Best of luck with the revisions

Reviewer

Majors –

 1) Introduction – you cover a lot of material and bring together lots of themes, which is important for this paper, however, I think it jumps around a bit and as a result I do not think it sets up your study as well as it could. While the description of the UGH was very clear and well evidenced, the GRH wasn’t given the same space. In particular, is this a hypothesis you have developed? If so this makes more sense, but I think you need to introduce it as such. Before mentioning the GRH you discuss the literature on kinship systems from an evolutionary perspective – I think what was missing for me was a link (not that I doubt it, just unclear as written) between kinship systems, ecologies, variance in RS and social networks. Is it the case you are arguing that different ecologies impact gender specific variance in RS which have consequences for kinship systems which will ultimately impact social networks. OR, are you arguing that the factors which impact the variation in kinship systems will independently impact social networks? Relatedly, I think the links with philopatry can be strengthened, why specifically is this important for social networks for instance (E.g. the relatedness argument). I think it’s a good point you make about human evolution but just needs to be more clearly tied up with your aims. Finally, while you discuss the importance of cooperation in the opening sentences, linking it up with human reproduction, I think it would be really important to discuss a bit about the function of social networks and cooperation, and how this varies by gender. This relates to my points above that why would social networks change in different settings, is it just a function of the different social composition of camps, or is it that women have different roles and thus different needs for social networks which will facilitate cooperation? You touch on this in the discussion but I think it really needs pulling out into the set up. To summarise, I think social networks need to be teased apart from kinship systems and cooperation, and a clearly perspective on what impacts what and why.

2) methods collection – great description of the community you work with however I felt more information was required for data collection and analysis. In particular, as you interviewed only one member of the household (right?) in the villages they would be answering for both the genders? I think the ‘instrumental’ social networks where at the household level, not the individual, and so cannot be separated into gender. However, the ‘hangout’ measure was – so what this asked specifically to both men and women in the household or just whoever responded? If whoever respondent, then were there systematic differences between who responded based on if a matrilineal village or a patrilineal village?

3) methods analysis – the variables household distance and reciprocity are not previously mentioned. How are these quantified and how do they relate to your predictions? Also, I assume the instrumental networks are childcare, household etc but this could do with defining beforehand. Relatedly, it is really important to include more information about why you picked those specific SN measures (degree and density), what they mean in terms of social network composition and how they related to your predictions. Also, more information about the scale for these measures would be helpful (e.g. 0-1). So it is clear that higher degree means more connections and larger social networks, relating directly to your directions but what about density? Further, it was unclear to me if this was a weighted network or not so perhaps this isn’t possible but wouldn’t degree vs. strength (e.g. the strength of weak ties literature) be a good way of capturing the large network of males vs. intimate and dyadic network of females assumptions? Or perhaps eigenvector centrality when you have a measure of how well connected are others in the social network?  I think increasing the discussion here will be really important.

Your analysis comprises of descriptive stats and ERGM. For the descriptive stats am I right in thinking you are using the global network for the given network? E.g. how do all the women’s friendships link up? So by doing so you only have two networks, broken down into hangout, childcare meals etc; one for matriliny and one for patriliny. As a result, you can only describe the differences between them. This is informative, and here you are saying as a collective, women’s networks have more ties and a higher density of ties than males global hangout network in matrilineal settings as compared to patrilineal settings. But how does this compare to ego specific networks? Do individual women have higher density and degree compared to individual men? And can this be predicted by kinship systems? My point is two-fold, is the wider research and your point about global or ego networks? I think this needs to be consider and discussed. My other point is that if you were to conduct the analysis at the ego level you would then be able to use inferential statistics to capture the variance around the means and thus some confidence about the degree of differentiation. I have no issue with the descriptives (the figures 1-2 were missing from my version, but found the pre-print online and found the distributions very informative) but I think could be strengthened. As I mentioned above, if I have misunderstood something and this analysis is not in fact possible then that is fine. The result tables for the ERGM (for someone who has never conducted them before) require further detail. What are the edges in this context? Is this the intercept? What is MCMC%? And were all the networks ran in the same model? Does this lead to any concerns about covariance between predictors? Finally, is it possible to run interaction terms in ERGM? To me it seems as you want to say the relationship between childcare and kinship (for instance) is moderated by whether it’s in a matrilineal or patrilineal settings. Therefore, to be able to capture this an interaction term is required. Finally, why was reciprocity missing for childcare and community meals? Did it just not happen?

Minors

Abstract  – I found the start of the abstract focusing on HBE framework too long and not focused enough to get a clear idea of the study. I would give more space to the UGD hypothesis.      

Line 34: you day ‘many have gone so far to argue’ but just cite Kramer so it might be worth including another refer here like Hrdy (2009)

Line 40: include universally here – I would agree that men and women follow different strategies but the issue is with assuming this strategies will be constant.

Lines 63-67: I think this is an issue with the original hypothesis but stood out to me, Benenson (1990) seems to say that males have both larger, more diffuse networks but then fewer heterosexual friendships – this seems like a contradiction to me? Perhaps specify if what friends are, and what a ‘larger’ network would be comprised of?

Line 69-86: excellent review

Line 68 – be specific that the above predictions are form the UGD hypothesis – you start of the next paragraph with it but it could be made more explicit.

Line 74 – ‘university men’ does that mean students specific or any males for worked or where involved in the university?

Line 99: e.g. for human flexibility would be good here.

Line 104: you summarise that several lines of evidence suggest flexibility, however your above points are focused really on how philopatry cannot be assumed in human evolution, so I would expand the evidence of flexibility (or clarify) so your summary works.

Line 130: so do you predict less evidence of dyadic relationships in UGR?

Lines 150-151: you say that you speculated in a previous publication that the gendered differences in health arise from more limited social support for women in patriliny – which you test here. However, you do not, as I have read it, explore the amount of social support, and your hypothesis is about the difference between weak and strong, or lots of ties vs. dyadic ties not about the support which flows through them?

Line 167: replace do with capture

Line 175: what does it mean to double sample? And can you confirm what layers are (primarily thinking of people who aren’t familiar with SNA)

Line 203: is friendship the hangouts?

Line 206: figures missing

Lines 202-227: somewhat repetitive as the results of Table 1 are summarised at the start and end. I would suggest condensing.

Line 321-325: I think it would be good to clarify what were conclusions from previous studies and what were conclusions from this study.

Lines 329-333: this depends, it would be important to sepeate out the number of supporters from their actions and male and female roles in these settings. i.e. in patrilineal system do women do more childcare? What other tasks are they doing? I think just needs sentence with ethnographic detail so it links up.

Lines 365 – start of the conclusion is missing.

Round 2

Reviewer 3 Report

I thank the authors for their consideration of mine, and the other reviewers, comments in the first round of reviews. I do find the paper greatly improved as a result. In particular, the authors have done an excellent job in bringing together a diverse range of perspectives in the introduction and addressing the reviewers’ methodological concerns. I do have a couple of minor comments which I think would be good to address, as I think it would strengthen the paper, however I am a n = 1 and if the other reviewers and editors do not think it is required then that is fine.

Introduction:

You cover lots of information, leveraging on diverse sources as evidence for your points, which is great. However, the section when reading feels rather long due mainly I think to longer sentences (E.g. lines 380-384). I think it would be useful to go through and cut-down some of the wordiness or more repeat statements. For instance in lines 397-402 seems to make the same point about humans being flexible in residence, I see one set of references for HG and then more widely, but it might be more focused to condense these into one sentence.

the last paragraph in the introduction is very clear and makes it clear how you are testing the UGD and GRH. However, because this is so explicit and the other centrality measures not really mentioned (I think) beyond the abstract I was somewhat less prepared for the descriptive results and the WRS/KS tests. Could a sentence just be added on, or brought to the fore from elsewhere, saying “UGD expects consistency in social networks by gender and we explore this with descriptives looking at four key centrality measures”. Or in the methodology you could say descriptive stats and WRS tests used to explore if male networks never change by context, if male/female networks do always differ, and we explicitly test the GRH with a GLM to see if females’ networks become more ‘male’ in a mat setting. I just think this might aid understanding.  

Methodology:

Excellently detailed in my opinion. Is it possible to here to detail what the centrality measures mean (e.g. what they are  measure and what 0-1 mean in each case) I see this is below in the results but I felt as the Table 1 came first it might be better placed in the methods.

Wilcoxon rank sum tests – I often think for smaller sample sizes and issues with data distribution permutation tests are preferable (See:  https://stats.stackexchange.com/questions/63863/what-is-the-benefit-of-using-permutation-tests ). In my own experience they are easy to run and interpret without being concerned of the distributions. In these, you compare your sample means to the simulated means, and simply ask out of 10,000 (or however many you want) simulations of your data how many cases you were expect to have a higher/lower value than you found. This proportion becomes the p-value. I will leave this up to you, but happy to send over code if of interest!

GLM – how you set up your research question it is clear that the interaction (which as you report is non-significant) is important to test your question. Again, this may just be me, but I would prefer to see the interaction model in the table, with the main effects of mat women and pat women (or mat women/mat man). The predicted probabilities are really useful, but it would be nice to see the B, SE and 95% CIs for these terms as well (also more generally can you include 95% CI?). This is just because I agree with you that the predicted probabilities show/hint at reversal and I think this information would be informative. You can get those main effects by just inverting the reference category and re-running the model (you can also just add the beta of the main effect and interaction, but by inverting the reference category you also get the SE, p and 95% which is helpful!).

Discussion:

Very well written, again tending on the long side but I do appreciate the importance of the ethnographic detail and discussion of the various elements here.

I think this is to address the other reviewers comments, whoever, I found the last sentences of the discussion a bit too negative (lines 1901-1904). I think you have already throughout the paper emphasised the limitations of the paper and addressed them as well as possible given constraints etc. I just felt this could be written more positively (e.g. strength is comparative nature and testing these core assumption, hope that others/future research can take this forward).

Some line specific comments:

Line 475-479: this point starts off with mobility but then the example doesn’t tie up well. Slight re-word.

Line 486-489: I think the main point is implicit here (so it must be the women who trade right?) and could be brought to the fore.

Line 875: what happened to the missing information – deleted observations?

Line 877-879: difficult to read.

Line 1354: I didn’t follow this sentence.

Lines 1356-1366: repeated paragraph from above.

All the best with the revisions

Author Response

Our comments are in italics. Please note that we do not reply to the minor criticisms point-by-point, but have done our best to address them in the revision. Thank you (again) for your helpful review!

I thank the authors for their consideration of mine, and the other reviewers, comments in the first round of reviews. I do find the paper greatly improved as a result. In particular, the authors have done an excellent job in bringing together a diverse range of perspectives in the introduction and addressing the reviewers’ methodological concerns. I do have a couple of minor comments which I think would be good to address, as I think it would strengthen the paper, however I am a n = 1 and if the other reviewers and editors do not think it is required then that is fine.

Introduction:

You cover lots of information, leveraging on diverse sources as evidence for your points, which is great. However, the section when reading feels rather long due mainly I think to longer sentences (E.g. lines 380-384). I think it would be useful to go through and cut-down some of the wordiness or more repeat statements. For instance in lines 397-402 seems to make the same point about humans being flexible in residence, I see one set of references for HG and then more widely, but it might be more focused to condense these into one sentence.

Thank you. We agree that this section is lengthy and have done our best to trim for readability. However, in order to adequately address all reviewers' concerns, and to present a thorough literature review, we have opted to retain most of the material.

the last paragraph in the introduction is very clear and makes it clear how you are testing the UGD and GRH. However, because this is so explicit and the other centrality measures not really mentioned (I think) beyond the abstract I was somewhat less prepared for the descriptive results and the WRS/KS tests. Could a sentence just be added on, or brought to the fore from elsewhere, saying “UGD expects consistency in social networks by gender and we explore this with descriptives looking at four key centrality measures”. Or in the methodology you could say descriptive stats and WRS tests used to explore if male networks never change by context, if male/female networks do always differ, and we explicitly test the GRH with a GLM to see if females’ networks become more ‘male’ in a mat setting. I just think this might aid understanding.  

Great point. We added a sentence at the end of the introduction to prepare readers for subsequent material.

Methodology: 

Excellently detailed in my opinion. Is it possible to here to detail what the centrality measures mean (e.g. what they are  measure and what 0-1 mean in each case) I see this is below in the results but I felt as the Table 1 came first it might be better placed in the methods.

We have added explanations of each measure to the Methods section.

Wilcoxon rank sum tests – I often think for smaller sample sizes and issues with data distribution permutation tests are preferable (See:  https://stats.stackexchange.com/questions/63863/what-is-the-benefit-of-using-permutation-tests ). In my own experience they are easy to run and interpret without being concerned of the distributions. In these, you compare your sample means to the simulated means, and simply ask out of 10,000 (or however many you want) simulations of your data how many cases you were expect to have a higher/lower value than you found. This proportion becomes the p-value. I will leave this up to you, but happy to send over code if of interest!

Thank you. We added permutation tests and they are indeed more sensitive. We have adjusted the results accordingly.

GLM – how you set up your research question it is clear that the interaction (which as you report is non-significant) is important to test your question. Again, this may just be me, but I would prefer to see the interaction model in the table, with the main effects of mat women and pat women (or mat women/mat man). The predicted probabilities are really useful, but it would be nice to see the B, SE and 95% CIs for these terms as well (also more generally can you include 95% CI?). This is just because I agree with you that the predicted probabilities show/hint at reversal and I think this information would be informative. You can get those main effects by just inverting the reference category and re-running the model (you can also just add the beta of the main effect and interaction, but by inverting the reference category you also get the SE, p and 95% which is helpful!).

We have added the model with the interaction term to the table and the text includes an explanation of the interaction (the effect of being a man in matriliny and patriliny, respectively). The corresponding figure has also been updated accordingly (to draw on the model with the interaction term).

Discussion:

Very well written, again tending on the long side but I do appreciate the importance of the ethnographic detail and discussion of the various elements here.

I think this is to address the other reviewers comments, whoever, I found the last sentences of the discussion a bit too negative (lines 1901-1904). I think you have already throughout the paper emphasised the limitations of the paper and addressed them as well as possible given constraints etc. I just felt this could be written more positively (e.g. strength is comparative nature and testing these core assumption, hope that others/future research can take this forward).

Thank you. As you note, we were perhaps overly cautious in the revision. We've modified the last sentence to be less negative and more future-facing.

Some line specific comments:

Line 475-479: this point starts off with mobility but then the example doesn’t tie up well. Slight re-word.

Line 486-489: I think the main point is implicit here (so it must be the women who trade right?) and could be brought to the fore.

Line 875: what happened to the missing information – deleted observations?

Line 877-879: difficult to read.

Line 1354: I didn’t follow this sentence.

Lines 1356-1366: repeated paragraph from above.

All the best with the revisions